# "…this could be a noble idea and a game changer." The potential of a dual prevention pill for HIV and pregnancy prevention among adolescent girls and young women in Zimbabwe

Adlight Dandadzi[1], Sanyukta Mathur[2], Petina Musara[1], Irene Bruce[3]*, Lorna Begg[3], Natasha T. Sedze[1], Prisca Mutero[1], Nyaradzo M. Mgodi[1], Barbara A. Friedland[3]

**1** University of Zimbabwe - Clinical Trials Research Centre, Harare, Zimbabwe, **2** Social and Behavioral Science Research, Population Council, Washington, District of Columbia, United States of America, **3** Center for Biomedical Research, Population Council, New York, New York, United States of America

* ibruce@popcouncil.org

## Abstract

Adolescent girls and young women (AGYW) in Zimbabwe bear a double burden of HIV and unmet need despite the progress made in provision of family planning and oral pre-exposure prophylaxis (PrEP). We elicited opinions from AGYW and health care providers (HCPs) about a dual prevention pill (DPP) in development that combines oral contraceptives (OCs) with oral PrEP to simultaneously prevent unintended pregnancy and HIV, and potentially increase uptake and adherence to oral PrEP. We enrolled 44 participants (March-June 2021) and conducted 12 in-depth interviews with HCPs (nurses, counselors, social workers, pharmacists, clinicians) from public and private health facilities offering HIV and family planning services in Harare, and four focus group discussions (FGDs) with 32 AGYW who were current OC users, stratified by age (16–19, 20–24). The HCPs and AGYW welcomed the idea of the DPP. Both groups perceived the benefits of the DPP as lessening the burden of taking two separate pills and giving AGYW the option to protect themselves discreetly, potentially increasing adherence and ease of use. HCP's had favorable attitudes toward the DPP and highlighted that the DPP regimen is already similar to the OCs women are taking and would allow for regular menses and quick return to fertility. Despite concerns about potential side effects and lack of protection against sexually transmitted infections (STIs) other than HIV, HCPs noted the potential benefits of the DPP in reducing their workload and increasing the uptake of PrEP services. From the end user perspective, the DPP size, color, and packaging should be appealing and distinct from HIV medications to minimize stigma. Clear guidelines are deemed necessary for DPP service provision for adolescents, emphasizing the importance of tailored approaches for different age groups. Educating male partners and the broader community about the DPP could enhance its use among AGYW.

**Data availability statement:** Data is available upon request to "Knowledge Management Specialist" who can be contacted at Publications@popcouncil.org.

**Funding:** This research was made possible by the generous support of the National Institute of Mental Health of the National Institutes of Health (Award Number R34MH119982 to Principal Investigators BAF, NMM, SM). Additional funding support was provided by the Children's Investment Fund Foundation (CIFF) (# 1903-03681, #2106-06589 to Population Council). The content is solely the responsibility of the authors and does not represent the official views of the National Institutes of Health or CIFF. The funders had no role in study design, data collection and analysis, decision to publish, or preparation of the manuscript.

**Competing interests:** The authors have declared that no competing interests exist.

## Introduction

Zimbabwe has made significant progress in the response to AIDS over the last decade and is one of the few countries that has achieved the "95-95-95" targets (meaning 95% of people living with HIV know their HIV status, 95% of people who know their HIV status are receiving antiretroviral therapy, and 95% of people on antiretroviral therapy have achieved viral load suppression) [1]. Thanks to focused prevention programs, Zimbabwe successfully reduced new HIV infections by 78% between 2010 and 2022 [1]. In 2022, Zimbabwe repealed laws that criminalized HIV exposure, nondisclosure, and transmission, which were seen as major obstacles for the HIV response [1].

Despite these advancements, women and girls still face obstacles to protecting their sexual health. In Zimbabwe, 13.7% of women are living with HIV compared to 8.2% of men [2]. There is a high prevalence of intimate partner violence; 31.3% of women aged 15–19 and 29.78% of women aged 20–24 experience intimate partner violence [2]. The prevalence of forced sexual initiation is 14.7%, which has been linked to increased susceptibility to HIV [3,4]. Female sex workers in Zimbabwe are at particularly high risk of HIV infection, with an estimated 45% prevalence of HIV and higher incidence among younger women [2,5]. One study estimated that 70% of all new HIV infections in Zimbabwe are directly or indirectly attributable to transmission during sex work [6]. Qualitative assessments in Zimbabwe have found women face barriers in HIV prevention, as many women struggle to negotiate condom use with their male partners [7].

Daily oral pre-exposure prophylaxis (PrEP) is highly effective in the prevention of HIV infection. However, PrEP effectiveness is significantly reduced in young people under the age of 30 due to low adherence and high discontinuation [8]. There has been low uptake of PrEP in sub-Saharan Africa, with stigma (related to presumed promiscuity and perceived negative impacts on sexual relationships) often cited as a reason for nonuse [9–11]. Therefore, strategies are needed to increase PrEP use among adolescent girls and young women (AGYW) at high risk of HIV.

One option is to develop new technologies that can meet multiple health needs and better meet the preferences of end users. Data indicate that women prefer, and are more likely to use, a multipurpose prevention technology (MPT) that prevents HIV and pregnancy over one that prevents only HIV or pregnancy [12–15]. To address this need, the dual prevention pill (DPP), a daily oral pill containing the active pharmaceutical ingredients in oral PrEP and in a combined oral contraceptive (COC) is currently under development [16]. Individuals at high risk of HIV can safely take PrEP and COCs together with no significant drug interactions [17,18]. Further, because the DPP combines two drugs that are licensed and marketed in many countries in sub-Saharan Africa, combining PrEP with a COC is likely to be the fastest route to an approved contraceptive MPT [19].

We conducted a qualitative study with health care providers (HCPs) and AGYW to gain their perspectives and insights about the DPP. Zimbabwe presented an ideal context for this study, as oral contraceptives (OCs) are widely used as a method of

contraception. Among all women aged 15–49 using modern contraception, 28.2% use contraceptive pills [20] and the prevalence is even higher among married women (41%) [21]. We hypothesized that women and HCPs have experience with OCs and could provide useful insights on the potential acceptability of a daily use oral MPT. Our study presents data on the potential acceptability of novel user-centered and user-controlled MPT and highlights key considerations for product introduction.

## Methods

### Sample size and population

We implemented this study from March and June 2021 at the University of Zimbabwe Clinical Trials Research Centre (UZ-CTRC) Zengeza Clinical Research Site (CRS) located on the grounds of the Zengeza Municipality Clinic in Chitungwiza, Zimbabwe, a suburb 25 kms from Harare.

**Adolescent girls and young women.** We conducted four (4) focus group discussions (FGDs) with AGYW currently using OCs, stratified by age (2 FGDs with 16–19-year-olds and 2 FGDs with 20–24-year-olds). To be eligible, AGYW had to be aged 16–24 and using OCs (per self-report), and willing to be audio recorded as part of the group discussion. We requested parental consent/adolescent assent for sexually active 16–17-year-olds who were not emancipated in accordance with current national guidelines in Zimbabwe. We recruited participants who did not know each other to reduce the likelihood of biasing the results. The Community Advisory Board (CAB) helped identify venues from which to recruit potential eligible AGYW, including family planning and primary health clinics, and HIV counseling and testing centers near the study clinic. We used both active and passive recruitment methods. Active methods involved recruiters speaking directly with potential participants and inviting those interested to come to the study clinic to learn more about the study, partnering with youth centers and health clinics offering sexual and reproductive health services, word of mouth, and messaging through community leaders. Passive recruitment included leaving information about the study (fliers) in the clinic waiting rooms so that AGYW who come in to access sexual and reproductive health (SRH) services who see the flyers and are interested would come to the study clinic for more information.

**Health care providers.** We conducted twelve (12) in-depth interviews (IDIs) with HCPs who had prior experience with PrEP. We aimed to interview several cadres of health care providers from public and private facilities that offer HIV and family planning services, including family planning and general health clinics, HIV counseling and testing centers, and pharmacies. We sought appropriate permissions within the province, district, and health centers to conduct interviews with HCPs. HCPs were selected from the participating clinics based on availability and cadre. All HCPs were informed that participation was voluntary and that if they chose not to participate, it would not affect their employment.

### Data collection procedures

**AGYW.** AGYW who were potentially eligible and interested were invited to the clinic where the study was explained in detail. Each FGD participant was consented individually, in their preferred language (English or Shona), to minimize coercion, and provided written informed consent prior to study participation. After providing informed consent, participants responded to a brief quantitative questionnaire about demographics and OC use prior to the FGDs. Participants were told that they could use a pseudonym during the FGD to preserve their confidentiality and that all data from the study would be reported in the aggregate without any personal identifiers. FGDs were conducted in a private room at the study clinic. FGDs lasted 1.5 to two hours and were facilitated by a moderator and a notetaker, who observed and recorded interactions and nonverbal cues during the discussions.

At the outset of each FGD, the researchers described the DPP using a flip chart and a standardized script. The FGDs then followed an interview guide developed by the team covering the following topics: AGYW's perspectives on pregnancy risk and avoidance strategies; HIV risk perception and avoidance strategies; PrEP knowledge, attitudes, and perceptions;

OC practices; opinions about the DPP (participants were shown a photograph of the co-formulated tablet being developed); what needs to be in place for young women to be able to use the DPP (including influence of partners, parents, providers, peers). FGDs were conducted in Shona, the local language, audio-recorded, transcribed, and translated into English.

**HCPs.** In most cases, IDIs were conducted at the HCP's facility or in a mutually agreed upon off-site private location. HCPs were asked to provide written informed consent prior to starting the interview. HCPs were also introduced to the DPP via the same flip chart with additional details. IDIs followed a standardized interview guide including a discussion of norms and practices among AGYW related to pregnancy prevention, HIV prevention, knowledge and attitudes about PrEP, opinions about the DPP, what needs to be in place for AGYW to have access to and be able to use DPP, and the legal and ethical considerations for introducing the DPP in Zimbabwe. IDIs lasted approximately one hour and were conducted in English, audio recorded, and transcribed.

### Data analysis

Guided by the framework for DPP development and introduction [22], we explored AGYW and HCP perspectives on three sets of factors – product-related characteristics, service provision characteristics, and end user characteristics and product use circumstances – that together are likely to influence product acceptability and intent to use. We therefore conducted a thematic content analysis of the data based on the DPP framework. Coders from the Population Council and UZ-CTRC teams participated in the development of a code book, coding of the transcripts, and identifying themes. For quality control of the coding process, 10% of the transcripts were coded by more than one analyst and regular team meetings were held to discuss and resolve disagreement in coding. When there were no differences in the understanding of and the application of a code, convergent validity was considered to have been achieved [23]. After coding the data, we summarized findings related to attitudes and beliefs about the DPP, considerations for DPP service delivery, and the social ecology of DPP acceptability and adherence among individual users [24,25]. Following thematic content analysis, we used a constant comparative method to compare whether the same concept emerged within, between, and across FGDs and IDIs for questions that were similar [26,27]. At every step of the analytic process, findings were reviewed by the investigators to validate codes and categories, and identify themes.

### Inclusivity in global research

Additional information regarding the ethical, cultural, and scientific considerations specific to inclusivity in global research is included in the Supporting Information (S1 Checklist).

### Ethical approval

The study protocol, consent forms, interview guides, flip chart, demographic questionnaire, and recruitment materials (flyers, posters) were all approved by the Population Council Institutional Review Board (New York, NY) and the following groups in Zimbabwe: the Ethics Committee of the Municipality of Chitungwiza; the Joint Research Ethics Committee for the University of Zimbabwe, Faculty of Medicine and Health Sciences and Parirenyatwa Group of Hospitals (JREC); the Medical Research Council of Zimbabwe (MRCZ), and the Research Council of Zimbabwe (RCZ). All participants were compensated an equivalent of US$10 for their participation per local ethical guidelines.

### Results

A total of 44 participants took part in the study, with 32 AGYW participating in the focus group discussions (6–8 per group) and 12 HCPs in in-depth interviews. **Table 1** presents AGYW respondent characteristics. The average age of the AGYW was 19.5 years while HCPs had an average of 43.4 years. Most of the AGYW (69%) had completed their secondary education, half of the AGYW were married, and 56% had used COCs for a period of 13 months or more. Less than half (41%) had used PrEP prior to study participation.

PLOS Global Public Health

**Table 1. AGYW respondent characteristics (n = 32).**

| Average age | 19.5 years |
|---|---|
| 16–19 years | 17 (53%) |
| 20–24 years | 15 (47%) |
| Completed secondary school | 22 (69%) |
| Married | 16 (50%) |
| Have been using COCs for 13 + months | 18 (56%) |
| Have used PrEP before | 13 (41%) |

Table 2 presents HCP respondent characteristics. Average age of the HCPs was 43 years and half were female. Different categories of HCPs participated in the study and represented different types of health facilities and organizations within the community.

Fig 1 presents a summary of findings from the study and the product, service provision, and end user characteristics related to DPP acceptability.

### Product characteristics

**Single DPP lessens the burden of two separate pills.** Most of the HCPs felt that taking a single pill was easier than taking two separate pills, with some of the AGYW emphasizing the convenience associated with taking a single pill. One HCP gave an example of tuberculosis (TB) treatment before the introduction of the fixed dose regimen where patients would have to take many pills, causing them to forget some pills and defaulting on their treatment. Instead with the DPP, HCPs expressed that the chances of forgetting to take one pill are low, unlike taking two separate pills where one would need to remember to take another pill from another packet.

*"So, I think it's good because it will reduce the pill burden…I think it will improve adherence for most of our clients especially on pill burden because it will be one tablet which covers two aspects…." (Pharmacist)*

**DPP regimen is familiar, would allow for regular menses, and quick return to fertility.** The DPP is being developed with a 21/7 regimen, in that women take the DPP (combined OC and PrEP tablet for 21 days) and then a PrEP-only tablet for 7 days. HCPs noted that this regimen is the same as the OC regimen that women are already

**Table 2. Health care provider (HCP) respondent characteristics (n = 12).**

| Average age | 43.4 years |
|---|---|
| Number of female HCPs | 6 |
| Type of HCPs | |
| Clinicians | 4 |
| Nurses | 2 |
| Counselors | 3 |
| Social worker | 1 |
| Pharmacists | 2 |
| Type of facility/organization | |
| Local public health clinic | 3 |
| Local public hospital | 2 |
| NGO clinic | 5 |
| Ministry of health or social welfare | 2 |

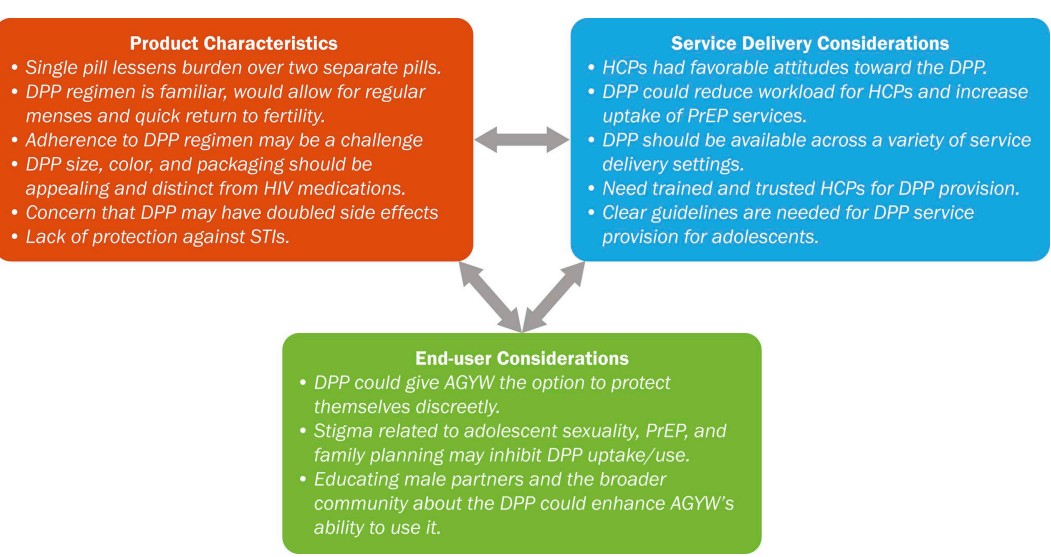

**Product Characteristics**
- Single pill lessens burden over two separate pills.
- DPP regimen is familiar, would allow for regular menses and quick return to fertility.
- Adherence to DPP regimen may be a challenge
- DPP size, color, and packaging should be appealing and distinct from HIV medications.
- Concern that DPP may have doubled side effects
- Lack of protection against STIs.

**Service Delivery Considerations**
- HCPs had favorable attitudes toward the DPP.
- DPP could reduce workload for HCPs and increase uptake of PrEP services.
- DPP should be available across a variety of service delivery settings.
- Need trained and trusted HCPs for DPP provision.
- Clear guidelines are needed for DPP service provision for adolescents.

**End-user Considerations**
- DPP could give AGYW the option to protect themselves discreetly.
- Stigma related to adolescent sexuality, PrEP, and family planning may inhibit DPP uptake/use.
- Educating male partners and the broader community about the DPP could enhance AGYW's ability to use it.

**Fig 1. Summary of Product, Service Delivery, and End User Considerations for the DPP.**

taking and would be acceptable. They also noted that the DPP regimen would allow women to "*flush out dirt [have their menses].*" However, other providers also mentioned that some women skip the placebo week to avoid their menses which could mean that the PrEP-only pills may not get used.

AGYW expressed considerable concern about delayed return of fertility upon discontinuation of long- acting hormonal contraceptives like Jadelle or Depo-Provera and other long-acting methods. Some of the AGYW mentioned that HCPs in their community only provide Jadelle to women with at least one child because of the aforementioned perception. On the other hand, participants noted that in Zimbabwe, OCs are widely used and seen as "*easily reversible.*" They felt that the DPP would be acceptable in this context because it would have a quick return to fertility.

> "*I think the pill is better than Jadelle because Jadelle you can have it for five years. Others say that if you are inserted Jadelle you might not get pregnant at all. So, the pill is good because when you just want to have a baby, when you just stop taking it, you can get pregnant.*" (AGYW, 16-19 age group)

**Adherence to DPP regimen might be a challenge**. HCPs noted that it might be cumbersome to use a daily pill for some and lead to adherence fatigue. They noted that a daily oral regimen might be challenging, especially for highly mobile women (like commercial sex workers, for example). AGYW were also concerned about taking the pill every day and that they might forget to take it given their busyness and the usual stressors of daily life.

> "*I see that the idea of taking it daily could be problematic for you to take it every day because at times you might forget, or you might be overwhelmed by events and then you become very busy or get stressed because of what will be going on and you forget to take your pill. Therefore, if it could be designed in such a way that it could be taken once per week, yah it will be better.*"
>
> (AGYW, 16–19 years)

> "*Because there are also adolescents who are in the key populations, the sex workers they will also have challenges. I guess because they do not stay at one place, today they are in Mazowe, tomorrow they are there, sometimes they leave their tablets and sometimes... so maybe this daily routine that's the only impact side [challenge].*" (SRH Clinician)

HCPs spoke of the daily-use regimen being associated with adherence fatigue. HCPs noted that often clients require additional counseling and support to sustain use of daily oral medications and avoid pill fatigue.

*"They get tired of taking medication at times, even if they know what is good for them but they do get tired at times that you [the provider] have to start afresh and counsel them about the importance of taking whether it is the ARVs or whatever the medication they are taking, so I hope they will not develop pill fatigue." (Nurse Counselor)*

AGYW and HCPs also noted that while DPP is a good start, there remained a need to diversify regimen (e.g., monthly pill) and delivery forms (e.g., providing dual prevention in the form of implants, intrauterine devices, vaginal rings, etc.), so that women who cannot take pills for different reasons are not disadvantaged. Participants felt that additional regimen and delivery forms would widen the options available, enhancing method choice for every woman to suit her preferences for HIV and pregnancy prevention.

*"Most say that they do not like taking a pill, they will want to be injected, those that come to the clinic. 'If only I was injected, drinking pills I do not like, then others say I do not like the pill I want to be injected.' So, when it is being manufactured, can't they manufacture a drug which is an injectable, because some opt to be injected, they do not want the oral route." (Counselor)*

**DPP size, color, and packaging should be appealing and distinct from HIV medications**. Participants reflected on the size, color, and packaging of the DPP and noted how the physical characteristics of the pill and its packaging could affect uptake and effective use. Participants noted that the size should be small for ease of swallowing. Additionally, AGYW wanted pills in an appealing or "appetizing" color.

*"If the pill is big, it becomes scary… Even to look at it because if something gets appetizing, it becomes attractive. If the pill becomes small, I take it that it is simple… But if it is big, you think, "What if I get choked." Even the paracet [paracetamol painkillers] themselves, they are scary to take. We do not like them. So, if it is smaller, you can take it freely rather than if it is big." (AGYW, 20-24 years)*

Both HCPs and AGYW noted that pill packaging needed careful consideration, as any association with packaging for antiretroviral (ARV) medications for HIV would lead to HIV-related stigma and influence uptake and use of the DPP.

*"So a concern with end users has been the packaging and how the tablets look… This came out quite… prominently with… PrEP tablets [that] they are looking like ARVs, which causes sort of a stereotype and even the way they will be in the same tin that contains ARVs and the size of the tablet… So, how the tablet is designed in terms of size and color and the packaging… try to make it as more on the contraception side and less on the ARV side. I think that will go a long way in like pushing away the stigma and normalizing the uptake of this." (Clinician)*

**Concern that DPP may have doubled side effects**. A common concern among both HCPs and AGYW is the potential for "dual side effects" with a DPP. The HCPs for instance, expressed how a person might experience diarrhea, skin rash, abdominal upset, sometimes changes in appetite because of PrEP. On the other hand, the COC might cause reduced libido, weight gain, heavy or light bleeding. Participants worried that the combination of the two medications might also mean a combination of their side effects and individuals' willingness to use the DPP.

*"…but I am concerned about maybe an exacerbation of the side effects so Truvada, this other tablet has nausea, vomiting, a bit of those headaches. So, if you…it's now being taken together with the estrogen-based [pills] which also have*

*the same side effects, I am a bit concerned that they might be an exacerbation of the side effects…it will also affect uptake." (Clinician)*

*"… the family planning pill could have side effects on you or the PrEP pill could have side effects on you and if the two are combined maybe it will be heavy for you…." (AGYW, 16-19 years)*

More specifically, both HCPs and AGYW noted that potential changes to menstrual bleeding patterns might lower willingness to use the DPP. AGYW noted that if the product led to an irregular menstrual cycle, this irregularity might create uncertainty and cause worry regarding their pregnancy status.

*"…you will be saying maybe this month you have a heavy blood flow and then maybe next month you skip, and the next month you produce some little blood. So, you will not know whether you are pregnant or not." (AGYW, 16-19 years)*

AGYW were also concerned about changes to their physical appearance – dark spots on the skin and changes in weight – both of which are associated in the community with individuals who are living with HIV. AGYW were worried that these potential side effects could expose them to stigma and discrimination.

*"…for sure when you move around in our community, because even if you develop a slim body,*

*[they will think] "You have AIDS." (AGYW, 20-24 years)*

AGYW were also concerned about rare but serious kidney problems associated with HIV medications. They worried about inadvertent disclosure under the custody of their parents. An AGYW participant presented the scenario below:

*"I bought it [DPP] from the pharmacy, it is the two of us with my pharmacist. My mother does not know this pill [DPP] that stays in my spare [bedroom] under the mattress. If I start developing kidney problems there, that would be difficult. Especially with my mother, she does not consider that you are a grown-up person, she will discipline you with a rod." (AGYW, 20-24 years)*

An HCP also flagged that there are existing myths and misconceptions of the side effects associated with hormonal contraceptives (such as its association with subfertility) that could influence DPP uptake.

With any potential side effects, AGYW noted that they needed to be fully informed/made aware of the benefits and side effects of the pill. HCPs concurred and felt that it would be important to weigh the benefits versus the risks. One HCP reiterated this sentiment by noting that the potential DPP side effects were minor as compared to having unwanted pregnancies and getting infected with HIV.

*"Yah side effects yah they are there, almost everything that we take has side effects. Of course, if we look at the side effects that I am seeing now, these are minor sometimes we can weigh out which is more advantageous, side effects or to get pregnant or to get HIV. Sometimes if you look at the side effects, people have developed them the first time and then after a long time they disappear. But even if it's stated they are not…it's not all women who are going to get those side effects." (Pharmacist)*

**Lack of protection against STIs**. An additional challenge raised by HCPs about the DPP included the lack of protection against other STIs. HCPs thought this might create some confusion in the counseling and that additional attention would be needed to clarify the need to use condoms for STI prevention. They noted that future products should offer protection against other STIs, as well.

## Service delivery considerations

**HCPs had favorable attitudes toward the DPP**. HCPs noted that AGYW suffer the double burden of unwanted pregnancies and HIV infection, hence the DPP (when introduced) will enable them to prevent both HIV infection and unintended pregnancies.

*"Yes, they will not have unwanted pregnancies, because these kids sometimes might get pregnant for people who will not marry them so it might – the pill is good for them because they can continue with their lives and their education without any fears. They will have no fears because they will be having something to protect them." (Counselor)*

HCPs felt that, given the lack of consistency in condom use, DPP offered a discreet option to protection from the dual burdens of unintended pregnancy and HIV risk.

*"Most of the times the consistency in terms of condom use has not been there, and it has led to many unwanted pregnancies and the transmission of HIV. So, when it comes to this pill where there is dualization in terms of contraception and HIV prevention, it means women on their own they are empowered. Just being able to take the tablet in the absence of men [without men's approval], it may be a success…this is a tablet which women can take on their own. It is just like the usual contraception but also with an added advantage of HIV prevention…. I think this could be a noble idea and a game changer." (Clinician)*

**DPP could reduce workload for HCPs and increase uptake of PrEP services**. HCPs noted that the DPP has the potential to reduce the burden on providers and facilities that offer HIV prevention and family planning services.

*"I think it would smoothen our flow of activities because currently we are giving family planning already. So, if we have something that is a two-in-one then it would mean that we have less burden and less time." (Clinician)*

Additionally, the HCPs expressed that the DPP could increase PrEP uptake at their facilities. They reported that PrEP uptake is currently low, and they were optimistic that the DPP would broaden HIV prevention efforts thereby increasing PrEP uptake as well as contraceptive uptake.

*"It might actually increase its uptake among adolescent girls and young women because those two issues are of major concern to them, pregnancy and HIV acquisition. So having those two issues addressed by one pill is…I think it's a good thing. It will actually increase uptake." (Clinician)*

**Need trained and trusted HCPs for DPP provision**. In terms of who should provide the DPP, health care providers emphasized the importance of appropriate training since the DPP contains an ARV. They felt that nurses, doctors, counselors, and pharmacists would be qualified to provide the DPP as they would be able to understand how the product works, the potential side effects, and how to counsel clients to make an informed decision. Some HCPs noted that community-based distributors (CBDs) would not be qualified as they did not have appropriate training, however others felt that community health workers could also play a pivotal role distributing the DPP because they have greater reach.

*"This pill should not be given by just a mere person, but it should be given by people who would have been taught about the DPP, how it works…. It should be given by people who would have been trained." (Counselor)*

*"But as we roll out and we are targeting more and more clients, I think the issue of task-shifting becomes very, very important. So the village health worker or the community nurse … they are not highly qualified and skilled, less*

*specialized but they have greater reach…. It has a greater reach than just waiting for clients to be coming to the facilities." (Clinician)*

This sentiment that village health workers could have more reach was also supported by a young woman who noted that village health workers could play a key role in spreading information about the DPP within the community.
Some AGYW expressed fears that providers themselves could spread misinformation.

*"…if they hear your years and you will be saying you want to collect contraceptive pills. Some nurses might tell you bad-bad ideas to say, 'if you start taking those pills you will be infertile permanently and so forth," that could prevent you [from using the DPP]. You will leave that place in confusion." (AGYW, 16-19 years)*

AGYW explained that certain providers may be better equipped to instill trust in patients. Some participants noted that frontline health care workers need to have the right information because that's where people go first. Many AGYW felt they would be more comfortable talking to younger health care workers, and those who were female, about the DPP.

*"…with someone of your age, you can disclose things that are deep down in your heart. But as for the older staff, for me to tell them that, [laughter] the grey-haired ones, that 'Aunt, I want DPP,' it even embarrasses me to say, 'Should I tell this old woman, who is my mother's age?'" (AGYW, 20-24 years)*

**DPP should be available across a variety of service delivery settings**. HCPs and AGYW emphasized the need for the DPP to be available at different types of health facilities in both the public and private sectors. HCPs explained that the population is diverse with diverse preferences for where they would like to receive services. Therefore, to maximize the impact of the DPP, it should be broadly available. **Table 3** summarizes the potential advantages and disadvantages of different service delivery settings for the DPP with illustrative quotes from HCPs and AGYW. The locations where DPP could be provided are noted in the order of preference noted by the participants. Family planning or HIV clinics were most frequently suggested as appropriate venues for DPP provision, though participants noted the privacy offered by venues like pharmacies and the counseling/support offered via community-based organizations.

**Clear guidelines are needed for DPP service provision for adolescents**. Many HCPs described the existing challenges providing SRH services to adolescents under the age of 18 years in Zimbabwe, noting "*those issues that are coming from family planning as ethical issues will also affect DPP."* HCPs pointed to the lack of harmony between the constitutional definition of a child (anyone under 18 years) and age of sexual consent laws (16 years). Further, legally HCPs are not allowed to provide services without parental consent. This lack of consistency puts HCPs in a tricky position, leaving each HCP to individually decide whether or not to offer SRH services to adolescents; they have to balance the laws with the adolescent's right to services.

*"I think it's something that we ought to discuss, maybe at policy level, to say someone has already given birth. Yes, they gave birth at 12, now they are 15, and they want this method [DPP]. So, for the health worker it's a fix because their rules are saying don't dispense if someone is below 16, they [the client] cannot access that [method]." (Clinician)*

HCPs differed on their stances on needing parental consent for the DPP, with some expressing that parental consent would be necessary and important, whereas others noted that adolescents who were not living at home, or who had already given birth, would not require the same level of parental involvement.

*"If they develop side effects, it's the parent who is going to take them for treatment, so they will ask…I am sure they would want to know why you gave them [DPP] without their consent." (Nurse Counselor)*

**Table 3. Potential DPP delivery location and advantages and disadvantages of each.**

| Location | Advantages | Disadvantages | Illustrative Quotes |
|---|---|---|---|
| Clinics (family planning or HIV prevention) | • Already offer PrEP and/or family planning services and many women are accustomed to going to clinics for care<br>• Easier to incorporate new services into places with existing services<br>• Provision of DPP at clinics will increase the number of clients coming for services – could increase the number of women using PrEP<br>• Availability of proper storage of medications to ensure product quality and effectiveness are not compromised<br>• Doctors/medical staff can clearly explain how to use the DPP<br>• Easy access, low cost, and consistent supply | • Negative attitude of health care providers towards AGYW/AGYW use of DPP<br>• Lack of privacy/confidentiality, especially for AGYW<br>• Shortage of counselors, who women are more comfortable with (vs nurses)<br>• Provision of DPP will create more work and more understaffing that is already an issue | *"It is better for it to be found at the clinic because the medicines will be safe…and at the clinic there will be a certain pharmacy that will already be there where all the medications will be stored."*<br>*(AGYW, 16–19 years)*<br>*"...in clinics, there are very few nurses who can talk to people nicely…" (AGYW, 20-24 years)*<br>*"Most adolescents will not feel free or comfortable to walk into a facility and go straight to the family planning room. They fear that maybe a next-door neighbor lady will see me or maybe a lady from church. You know they still, they still fear all that." (SRH Clinician)* |
| Hospitals | • Trustworthy, doctors would be able to clearly explain how to use it<br>• Cheaper than at a pharmacy<br>• It could be a point of access once people are already taking it | • AGYW experience embarrassment going to hospitals, not an ideal setting to introduce women to the DPP. However, women go there anyway. | *"…if it is sold at the hospital, it will be cheaper than at the pharmacy. They would want more money at the pharmacy than at the hospital. We are shy to go to the hospital, but we just go anyway."*<br>*(AGYW, 16–19 years)* |
| Pharmacy | • Would provide much-needed privacy<br>• No personal information is collected (unlike public clinics)<br>• Less stigma/fear of being judged<br>• Some people already get family planning methods at pharmacies and are used to getting it there | • Likely to cost more at pharmacy than at public facilities<br>• May not have skilled providers to explain how to take the DPP, potential risks, etc.<br>• Limited availability of pharmacies in rural areas | *"… I am used to go to the pharmacy, to buy from the pharmacy because that is where I see it is easy for me."*<br>*(AGYW, 16–19 years)*<br>*"…let it [DPP] be available in pharmacies, so that people out here can be helped, because if it is in hospitals, in hospitals they will ask us our ages, to say, 'How old are you, why do you want to use it.'"*<br>*(AGYW, 20–24 years)* |
| Community-based organizations | • Adolescent-friendly; safe space for adolescents<br>• Clients are comfortable with community-based service providers – they are known, not strangers | • May be more difficult to follow up with clients<br>• Insufficient supply of appropriately trained providers | *"This DPP, which could be available, if they go to the youth center…they will be taught comfortably without feeling pressured."*<br>*(Counselor)* |
| Mobile vans | • Easier access than clinics – less travel time/money<br>• Already used for vaccination programs<br>• Can provide services to a lot of people in a community | • Must not be branded to be discreet<br>• Influential during initial stage of rollout, but loss to follow-up is an issue<br>• Difficult to manage regular follow-up with necessary HIV testing to avoid resistance when taking PrEP | *"Mobile vans, they are okay. But what happens is, what I have seen that mostly happens, when a commodity or an item comes – whether it's a pill like a DPP or a service – in the inception it's done very nicely. We use vans, we use everything. But when the rollout stage is gone, it actually collapses." (Nurse)* |
| Schools | • It would be ideal because adolescents are engaging in sex<br>• Adolescents would not have to face the negative health care worker attitudes | • Providing the DPP at schools will be seen as encouraging adolescents to have sex<br>• Ministry of Education not keen for health care workers to come to provide services (hard to provide condoms and contraception in schools)<br>• Requires parental permission and parents don't want schools talking to their children about sex<br>• Lack of privacy | *"If it can be available in schools because the 16-year-olds are in schools, right?"—(AGYW, 20–24 years)*<br>*"…the Ministry of Primary and Secondary Education – they are not very keen for health care workers to come with contraceptives. It's very sensitive…it's not really acceptable. But it's something that is…I think it's an ongoing debate in parliament." (Clinician)* |

*(Continued)*

**Table 3.** (Continued)

| Location | Advantages | Disadvantages | Illustrative Quotes |
|---|---|---|---|
| Home-based provision | • Convenient<br>• Easily accessible | • Could put women in danger if in abusive relationships<br>• Stigma from community | *"Can you imagine I knock at your door and I say I have come to talk about family planning. Maybe the husband doesn't want to hear anything about family planning. Already you can see what you are facing." (Nurse Counselor)* |

HCPs yearned for proper and clear national guidelines about who should or should not get the DPP. They noted discomfort with the current situation of needing to make judgments on a case-by-case basis about providing contraception and other SRH services to adolescents without parental consent.

## Considerations for end users

**DPP could give AGYW the option to protect themselves discreetly.** One of the key benefits of the DPP perceived by the HCPs and end users includes the potential for discreet use. Respondents further explained that women would be able to protect themselves without having to rely on their male partners to agree to use condoms especially in situations where the male partners are not willing to disclose their HIV status.

*"Because if we say, let's use a condom, he will refuse. So, with this [DPP], he will never know…I will be just taking [the pill] saying I am preventing pregnancy...I would know that I am preventing the disease [HIV] as well. I will be safe like that." (AGYW, 20-24 years)*

The dual protection offered by the DPP could help women navigate difficult circumstances where the male partners do not approve of PrEP or OC use. For women whose male partners disapprove of PrEP use, they can avoid the topic of HIV prevention with the partner by saying the pill is just for family planning, whereas women with partners who are against the use of contraception can say it is only for HIV prevention.

*"Sometimes there are partners who are very difficult, they don't want you to take PrEP. So, it's easy to say I am taking family pills...I have gone to the clinic; I got the family planning pill for family planning and yet behind there is PrEP. There is not much squabble there, people will just tend to say it's okay. So, for those clients who have got problems like that, of that type, they actually get an advantage…. If the husband accepts PrEP and doesn't accept family planning, it means the family planning is an advantage because if you take DPP it means you are taking a family planning method. So sometimes it becomes an advantage to some couples who have got problems." (Nurse)*

AGYW also pointed out that the DPP will be useful to protect women from HIV if their husbands are having extramarital sex. In those situations, while PrEP use may not be feasible/cause distrust in relationships, the combination in the DPP would allow AGYW to use DPP discreetly.

*"Because if you are in the house [married] and you know that your husband cheats and he also know that he cheats, for him to see you taking PrEP-only tablets there will be a problem there. So, for you to take them combined, he won't be able to stop you from taking your PrEP." (AGYW, 16-19 years)*

**Stigma related to adolescent sexuality, PrEP, and family planning may inhibit DPP uptake/use.** HCPs noted that adolescents (those under 18 years of age) may fear retribution from their parents about being sexually active. Reflecting on their experiences with dispensing family planning to adolescents, an HCP noted:

  

*"But the adolescents, there is a problem because if the parent doesn't approve, it means she must go behind the scenes and get the drug [family planning]. And that one becomes a problem in terms of secrecy to say where did you get the drug, how are you going to take it if your parent doesn't approve it." (Nurse)*

AGYW also expressed that it would be difficult to tell their parents about being sexually active or wanting to use the DPP which could make it more difficult for them to access. Instead, AGYW noted the importance of having support from other trusted or close family members/friends who could advise on DPP use.

*"Aunt, I think there are some secrets that you cannot share with your mum, but you can find a sister or someone older than you. You can tell her to say, "Aah, dear, these are the things that I am what, that I am thinking." She can give you better advice, your best friend, or a close relative, they can support you than going to your mother. Because going to your mother for sure it is scary, and you cannot even start because even the words to introduce your story." (AGYW, 20-24 years)*

AGYW also expressed concerns about existing stigma associated with PrEP use, as well as lack of support around family planning use.

*"…someone might see you going to take your oral pills, oral PrEP, and they hear rattling of pills and they would start spreading that, "This one has a disease, I saw her carrying the pills," because they don't know, they are not aware that there is PrEP. So they will be thinking that you have a disease and they will start saying, "This one has the disease." They will be pointing at you in the community." (AGYW, 16-19 years)*

*"…even the churches they might say we don't allow family planning. That will be preached to you at the churches like the Morani [pseudonym of apostolic sect], they will be saying we don't use family planning." (AGYW, 20-24 years)*

**Educating male partners and the broader community about the DPP could enhance AGYW ability to use it.** While respondents touted the possibility of discreet use of the DPP, when asked about what would make the DPP successful, AGYW said that male partners and others in the community (including religious leaders) should be educated about the DPP.

AGYW noted that partners' awareness about the DPP would enhance their support for the product and enable use among women. One young woman noted that male partners may be unsupportive of their partner's using the DPP, as they may approach a new, unknown product with skepticism.

*"I might be forbidden by my husband… He will be saying, 'I do not yet have adequate information.'" (AGYW, 20-24 years)*

They suggested that if their male partners understood the DPP, they could play a role in supporting AGYW having access to the DPP.

*"It is true that our partners must be educated first. If they understand, I do not see any problem with women because us women are the ones who are at risk. It is us who know our wars [risk of HIV and unwanted pregnancies]." (AGYW, 20-24 years)*

Another young woman commented that men, if supportive of the DPP, can spread awareness of the DPP among their male peers and encourage other men to support their wives' use of the DPP.

*"Yes, they can tell them for instance when they would have gone to the clubs [nightclubs] and they tell their peers to say, 'Ah there is a pill that is available, if you have multiple partners, your wife will not get pregnant, she will not contract the disease. So, she must go and get it.' They can tell each other to say, 'Is your wife using this pill for prevention and so forth.'" (AGYW, 16-19 years)*

Several AGYW mentioned that awareness campaigns through road shows, social media, radio, and village health workers would be key to educating not only women but also other community members about the DPP.

*"Because if we do the road shows, everyone will be able to know that there is a pill that is being developed so you will not feel that embarrassed to use it because people would know that this pill prevents what – HIV and pregnancy. So you will not feel very embarrassed." (AGYW, 16-19 years)*

Additionally, AGYW emphasized the desire to hear accurate information about the DPP and if it works. Some pointed to the spread of misinformation through informal networks and social media during the COVID-19 pandemic. Respondents emphasized the need for information spreading through reliable sources and from women who had actually used the product.

*"At times, the pill will be available, but the other thing that can be a roadblock are the rumors that we speak. Since the time COVID started, people have been saying different things. Some words to scare each other and so forth can hinder us from taking that pill. So, rumors themselves, so we need to find where we can get the knowledge rather than hearing about it from my friend next door." (AGYW, 16-19 years)*

*"…someone who once used the pill, so people can know that it works..." (AGYW, 20-24 years)*

## Discussion

Our study sought to understand AGYW and health care provider perspectives about the DPP, a novel daily oral MPT for HIV and pregnancy prevention that is currently under development. Overall, we found that AGYW and HCPs like the idea of an MPT that combines HIV and pregnancy prevention. Interest in the DPP was mainly attributed to perceived primary benefits cited as lessening the burden of taking two separate pills, empowering women to protect themselves against HIV infections and unintended pregnancies, reducing burden for HCPs, and potentially increasing PrEP use. The HCPs, in particular, expressed that the DPP is highly acceptable because it enables them to respond to women's needs by expanding options and addressing multiple health needs. On the other hand, all participants highlighted potential challenges that might impede successful uptake and adherence if not well addressed and considered in the product development and implementation.

Our findings confirm prior end user studies that have indicated that there is a need for MPTs that prevent HIV and pregnancy [28]. Compared to single-indication products, discrete choice trials indicate that women interested in HIV prevention have a greater demand for MPTs [12]. In a research involving women in Kenya and South Africa between the ages of 18 and 30, it was discovered that the participants "overwhelmingly" preferred a combined HIV and pregnancy prevention product over two separate products, and the majority were willing to give up their preferred single-indication product (injection) in favor of a less preferred product form that provided dual protection [29]. Our study confirms this strong interest in a product that provides protection against two overlapping risks/vulnerabilities.

Both AGYW and HCPs appreciated the potential of a self-administered product that would allow AGYW to protect themselves. However, they had mixed opinions about some of the product characteristics. Participants liked that the DPP contained OC, a method preferred by clients and HCPs because it offered a quick return to fertility. At the same time, they

expressed concerns about the potential for the "dual side effects" with a DPP. Similarly, the daily oral regimen for the DPP was perceived both as an advantage and a source of concern. Participants noted that the DPP regimen was similar to the OC regimen, a familiar and an often used method of contraception by women in our study setting [30]. At the same time, participants pointed out that it might be cumbersome to use a daily pill and might be a particular challenge for highly mobile women. Some HCPs were worried about clients skipping week 4 of the DPP pack that contains oral PrEP pills as women often skip the week 4 (placebo week) for OCs to avoid their menses. They noted that for many women it will be new information to learn that the week 4 pills are not placebos and are necessary to continue protection against HIV. The HCPs recommended DPP-specific training with trusted providers on management of side effects and adherence-counseling messages to enable AGYW to make an informed decision. This finding aligns with the idea of creating counseling materials that place the DPP in the context of a variety of contraceptive methods and increasing options for HIV prevention to ensure shared decision-making between providers and users, for women to select the method that best matches their prevention priorities [22].

To support eventual uptake and use of the DPP, participants underscored several considerations related to the context in which the DPP is provided. First, there was a robust discussion around making the product more accessible once it is available. Public health clinics offered the easiest sites to integrate the provision of an MPT with available contraceptive and HIV services and associated service providers.However, provider attitudes and anticipated stigma often kept AGYW away. Participants suggested broader distribution points beyond the public health clinics to reach AGYW [22] who might benefit most from the DPP. Second, HCPs also stressed the need for clarity on statutory laws regarding SRH service provision to AGYW. HCPs revealed that a number of obstacles, such as age-based restrictions on the provision of SRH services, stand in the way of girls who are aware of the advantages of PrEP and contraception and would like to receive these services. Hence the government must enact laws and policies to guarantee that everyone, especially adolescent girls, has access to SRH services [30]. Third, participants pointed to several preexisting stigmas that would influence the intention to use DPP and DPP initiation, including stigmas related to adolescent sexuality, PrEP and HIV, and contraception. For instance, participants asserted the need to clearly distinguish DPP packaging and provision from HIV and PrEP, and to align it more with contraception packaging to avoid HIV/PrEP-related stigma [22,31]. Participants also noted that spreading awareness about the DPP is crucial for women's ability to effectively use the product. Broad-based awareness about the product, not only with HCPs, but also parents/guardians, male partners, key community leaders (e.g., religious leaders) would be needed to reduce misinformation and create a supportive environment for product use. Additional efforts may be needed to ensure that AGYW clients can access the DPP, including programs to address provider bias about DPP, contraception, and PrEP for AGYW, which have been shown to be a barrier to family planning and HIV service use among AGYW [32,33].

## Limitations

Our study is not without limitations. This is a small formative qualitative study conducted about a hypothetical product. While we described the likely characteristics of the DPP in detail to all participants, future research with the actual product, among women who have experience with the product, and among providers who have experience administering it are warranted. Our study only explored the perspectives of AGYW in one site in Zimbabwe; subsequent research will need to examine perceptions about the DPP among different age groups and across different contexts.

## Conclusion

We find that DPP may be an acceptable option for AGYW seeking HIV and pregnancy protection. As a woman-centered and woman-controlled option that prevents HIV and pregnancy, the DPP has the potential to increase the number of women who receive oral PrEP, enhance its successful use, and, thus, contribute to the reduction of HIV incidence and unplanned pregnancies. Participants cautioned on product characteristics and use considerations that require careful counseling and consideration to ensure that the DPP, once available, is delivered and used effectively.

## Supporting information

**S1 Checklist.  Inclusivity in global research statement.**
(PDF)

## Acknowledgments

The study team would like to thank the young women and health care providers who participated in this study. We acknowledge Tracy McClair, at Population Council, for her technical support in the development of the data collection instruments and initial data analysis. We also thank Tendai Mamvura, at the University of Zimbabwe-Clinical Trials Research Centre, for her support during the delivery of the project. We also thank Joyce Altman for her editorial review.

## Author contributions

**Conceptualization:** Sanyukta Mathur, Barbara A. Friedland.

**Formal analysis:** Adlight Dandadzi, Sanyukta Mathur, Petina Musara, Irene Bruce, Lorna Begg, Natasha T. Sedze, Prisca Mutero, Barbara A. Friedland.

**Investigation:** Adlight Dandadzi, Natasha T. Sedze, Prisca Mutero.

**Supervision:** Petina Musara, Nyaradzo M. Mgodi.

**Writing – original draft:** Adlight Dandadzi, Sanyukta Mathur.

**Writing – review & editing:** Adlight Dandadzi, Sanyukta Mathur, Petina Musara, Irene Bruce, Lorna Begg, Natasha T. Sedze, Prisca Mutero, Nyaradzo M. Mgodi, Barbara A. Friedland.

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
