## [Decision Letter · Decision Letter 0]

6 Sep 2024

PGPH-D-24-01238

“…this could be a noble idea and a game changer.” The potential of a dual prevention pill for HIV and pregnancy prevention among adolescent girls and young women in Zimbabwe

Dear Dr. Dandadzi,

Thank you for submitting your manuscript to PLOS Global Public Health. After careful consideration, we feel that it has merit but does not fully meet PLOS Global Public Health’s publication criteria as it currently stands. Therefore, we invite you to submit a revised version of the manuscript that addresses the points raised during the review process.

Please note that we have only been able to secure a single reviewer to assess your manuscript. We are issuing a decision on your manuscript at this point to prevent further delays in the evaluation of your manuscript. Please be aware that the editor who handles your revised manuscript might find it necessary to invite additional reviewers to assess this work once the revised manuscript is submitted. However, we will aim to proceed on the basis of this single review if possible. 

We look forward to receiving your revised manuscript.

Kind regards,

Vanessa Carels

Staff Editor

Journal Requirements:

1. You indicated that you had ethical approval for your study. In your Methods section, please ensure you have also stated whether you obtained consent from parents or guardians of the minors included in the study or whether the research ethics committee or IRB specifically waived the need for their consent.

2. Please include the following request in the decision letter, and ping me with follow-up. “Please include a complete copy of PLOS’ questionnaire on inclusivity in global research in your revised manuscript. Our policy for research in this area aims to improve transparency in the reporting of research performed outside of researchers’ own country or community. The policy applies to researchers who have travelled to a different country to conduct research, research with Indigenous populations or their lands, and research on cultural artefacts. The questionnaire can also be requested at the journal’s discretion for any other submissions, even if these conditions are not met.  Please find more information on the policy and a link to download a blank copy of the questionnaire here: https://journals.plos.org/globalpublichealth/s/best-practices-in-research-reporting. Please upload a completed version of your questionnaire as Supporting Information when you resubmit your manuscript.

Additional Editor Comments (if provided):

Reviewers' comments:

Reviewer's Responses to Questions

**Comments to the Author**

1. Does this manuscript meet PLOS Global Public Health’s publication criteria? Is the manuscript technically sound, and do the data support the conclusions? The manuscript must describe methodologically and ethically rigorous research with conclusions that are appropriately drawn based on the data presented.

Reviewer #1: Yes

2. Has the statistical analysis been performed appropriately and rigorously?

Reviewer #1: N/A

3. Have the authors made all data underlying the findings in their manuscript fully available (please refer to the Data Availability Statement at the start of the manuscript PDF file)?

Reviewer #1: Yes

4. Is the manuscript presented in an intelligible fashion and written in standard English?

Reviewer #1: Yes

5. Review Comments to the Author

Reviewer #1: This is a well written article evaluating the opinions of users (adolescent girls and young women) and providers (health care providers) in Zimbabwe on the introduction of a dual prevention pill (DPP). The study showed that the DPP was welcomed by both providers and users but there were some concerns about potential additive side effects, lack of protection against other STIs and needing to distinguish the packaging from antiretrovirals to minimize stigma.

Minor corrections

Abstract

It would be beneficial to include the sample size in the abstract so the readers can know upfront on the scale of the study

Introduction

Line 90-93 - this sentence, "Guided by the framework.... and intent to use" may belong better in the methods

Results

Table 1 - please include the range for the ages. Given that the group of AGYW were stratified into 2 age groups (16-19 years and 20-24 years), can you include the proportion < 20 and > 20 years in the table? Of the 32 AGYW who participated, it is not clear if this was equally divided between the 2 age strata

Line 186 - can you provide the % for the sentence "Most (??%) of the HCP..."

Discussion

Line 520-521 - please include a reference for the other end user studies that indicate there is a need for MPTs

Line 559-560 - please rephrase the sentence. Something seems to be missing from the sentence

References

Please provide a doi for ref 6, 7, 22, 27 and 33

ref 17 - provide date accessed

Ref 18 - there seems to be newer guidelines published in 2024 - MMWR 2024; 73(4);1–126 - if appropriate, the newer guidelines should be referenced. if the 2020 publication is still appropriate, provide the end pages, i.e., 405-410

Ref 26 - provide publisher and place pf publication

Ref 28 - provide date accessed

Overall this study makes a useful contribution to the field of multipurpose prevention technologies.

6. PLOS authors have the option to publish the peer review history of their article (what does this mean?). If published, this will include your full peer review and any attached files.

**Do you want your identity to be public for this peer review?** For information about this choice, including consent withdrawal, please see our Privacy Policy.

Reviewer #1: No

---

## [Decision Letter · Decision Letter 1]

15 Apr 2025

PGPH-D-24-01238R1

“…this could be a noble idea and a game changer.” The potential of a dual prevention pill for HIV and pregnancy prevention among adolescent girls and young women in Zimbabwe

Dear Dr. Bruce,

Thank you for submitting your manuscript to PLOS Global Public Health. After careful consideration, we feel that it has merit but does not fully meet PLOS Global Public Health’s publication criteria as it currently stands. Therefore, we invite you to submit a revised version of the manuscript that addresses the points raised during the review process.

There are some very minor comments from the reviewer to address, can you please address these in the next revision? We hope these aren't too complex to consider, and thank you in advance.

We look forward to receiving your revised manuscript.

Kind regards,

Julia Robinson

Executive Editor

Journal Requirements:

Additional Editor Comments (if provided):

Reviewers' comments:

Reviewer's Responses to Questions

**Comments to the Author**

1. If the authors have adequately addressed your comments raised in a previous round of review and you feel that this manuscript is now acceptable for publication, you may indicate that here to bypass the “Comments to the Author” section, enter your conflict of interest statement in the “Confidential to Editor” section, and submit your "Accept" recommendation.

Reviewer #2: (No Response)

2. Does this manuscript meet PLOS Global Public Health’s publication criteria? Is the manuscript technically sound, and do the data support the conclusions? The manuscript must describe methodologically and ethically rigorous research with conclusions that are appropriately drawn based on the data presented.

Reviewer #2: Yes

3. Has the statistical analysis been performed appropriately and rigorously?

Reviewer #2: N/A

4. Have the authors made all data underlying the findings in their manuscript fully available (please refer to the Data Availability Statement at the start of the manuscript PDF file)?

Reviewer #2: Yes

5. Is the manuscript presented in an intelligible fashion and written in standard English?

Reviewer #2: Yes

6. Review Comments to the Author

Reviewer #2: Thank you for the opportunity to review the manuscript, “…this could be a noble idea and a game changer.” The potential of a dual prevention pill for HIV and pregnancy prevention among adolescent girls and young women in Zimbabwe”. The paper is well-written and presented, and I have only minor comments.

- Why were IDIs not done with the end users? Could the authors give some rationale why only FGDs were done for end-users and IDIs for HCPs?

- How many participants per FGD on average? I think the authors give only aggregate number (32 for total 4 FGDs).

- Line 189: Was there a quantitative survey? Or is the percentage based on the contents from IDIs? I wouldn't put percentage, if these data from IDIs.

7. PLOS authors have the option to publish the peer review history of their article (what does this mean?). If published, this will include your full peer review and any attached files.

**Do you want your identity to be public for this peer review?** For information about this choice, including consent withdrawal, please see our Privacy Policy.

Reviewer #2: No

---

## [Editor Report · Decision Letter 2]

31 Jul 2025

“…this could be a noble idea and a game changer.” The potential of a dual prevention pill for HIV and pregnancy prevention among adolescent girls and young women in Zimbabwe

PGPH-D-24-01238R2

Dear Miss Bruce,

We are pleased to inform you that your manuscript '“…this could be a noble idea and a game changer.” The potential of a dual prevention pill for HIV and pregnancy prevention among adolescent girls and young women in Zimbabwe' has been provisionally accepted for publication in PLOS Global Public Health.

Best regards,

Julia Robinson

Executive Editor